# 2D MXene Nanosheets with ROS Scavenging Ability Effectively Delay Osteoarthritis Progression

**DOI:** 10.3390/nano14191572

**Published:** 2024-09-29

**Authors:** Hongqi Zhao, Tianqi Wang, Xuan Fang, Tao Xu, Jian Li, Shaoze Jing, Guangzi Chen, Yang Liu, Gaohong Sheng

**Affiliations:** 1Department of Orthopedics, Tongji Hospital, Tongji Medical College, Huazhong University of Science and Technology, Wuhan 430030, China; 2Third Hospital of Shanxi Medical University, Shanxi Bethune Hospital, Shanxi Academy of Medical Sciences Tongji Shanxi Hospital, Taiyuan 030032, China

**Keywords:** MXene, Ti_3_C_2_T_x_ nanosheets, osteoarthritis, reactive oxygen species scavenging, articular cartilage, two-dimensional material

## Abstract

MXenes nanosheets with high conductivity, hydrophilicity, and excellent reactive oxygen species (ROS) scavenging ability have shown promise in treating various degenerative diseases correlated with abnormal ROS accumulation. Herein, the therapeutic potential of Ti_3_C_2_T_x_ nanosheets, which is the most widely investigated MXene material, in delaying osteoarthritis (OA) progression is demonstrated. In vitro experiments indicate the strong ROS scavenging capacity of Ti_3_C_2_T_x_ nanosheets and their acceptable biocompatibility. Ti_3_C_2_T_x_ nanosheets effectively protect chondrocytes from cell death induced by oxidative stress. In addition, Ti_3_C_2_T_x_ nanosheets demonstrate a prominent anti-inflammatory effect and the ability to restore homeostasis between anabolic activities and catabolic activities in chondrocytes. Furthermore, RNA sequencing reveals the potential mechanism underlying the Ti_3_C_2_T_x_ nanosheet-mediated therapeutic effect. Finally, the in vivo curative effect of Ti_3_C_2_T_x_ nanosheets is verified using a rat OA model. Histological staining and immunohistochemical analyses indicate that Ti_3_C_2_T_x_ nanosheets effectively ameliorate OA progression. Conclusively, the in vitro and in vivo experiments suggest that Ti_3_C_2_T_x_ nanosheets could be a promising and effective option for OA treatment.

## 1. Introduction

Osteoarthritis (OA) is a prevalent joint disease characterized by progressive cartilage degradation and chronic low-grade inflammation [1,2]. It is estimated that OA affects over 300 million people and results in a financial burden of 460 billion dollars [3,4]. Current management strategies aim to alleviate pain, improve joint function, and slow disease progression through a combination of pharmacological interventions, lifestyle modifications, and physical therapy [5]. However, these strategies fail to halt OA progression and most patients have to undergo surgical interventions such as joint replacement when OA progresses to the end-stage [6].

Biomaterials offer a multifaceted approach to OA treatment by addressing various aspects of joint pathology. For example, hydrogel can be engineered to have the ability to form a tenacious hydration layer, providing hydration lubrication and protecting the articular cartilage from mechanical damage [7,8,9]. Furthermore, biomaterials can be intra-articular injected and serve as carriers for bioactive molecules such as growth factors, cytokines, and anti-inflammatory agents, enabling targeted drug delivery to the affected joint while minimizing systemic side effects [10,11,12]. Additionally, the versatility of biomaterials allows for the integration of gene therapy and nanomedicine into OA treatment strategies [13]. Hence, delving into the potential of novel biomaterials in treating OA may provide a fresh perspective on developing effective OA management strategies.

MXene, a class of two-dimensional (2D) transition metal carbides, nitrides, and carbonitrides, has a wide range of applications across various fields owing to its attractive properties like high conductivity, hydrophilicity, redox-active surfaces, and tunable surface functional groups [14,15]. Similarly, MXene has emerged as a promising biomaterial in various biomedical applications [16,17]. Firstly, MXene shows excellent biocompatibility. Furthermore, the high surface area of MXene nanosheets endows them with high drug-loading capacity and makes them a promising candidate in drug delivery applications [18,19]. MXene also exhibits broad-spectrum reactive oxygen species (ROS) scavenging capacity. Feng et al. indicated that a 2D vanadium carbide could restore redox homeostasis and alleviate inflammatory and neurodegenerative diseases [20]. Zhao et al. demonstrated that Ti_3_C_2_T_x_ nanosheets could serve as antioxidant platforms for scavenging the overproduced ROS resulting from acute kidney injury [21]. In addition, MXene shows great osteogenic potential, pro-angiogenic ability, and extraordinary anti-bacterial effect [22]. Conclusively, MXene is a versatile and biocompatible 2D biomaterial and shows great promise in disease treatment.

Excess ROS could be produced when articular cartilage is exposed to overloaded mechanical pressure and contributes to the degeneration of articular cartilage [23,24]. It is well-established that biomaterials capable of scavenging ROS could ameliorate OA progression [25,26]. MXene may possess therapeutical potential in alleviating OA as it has excellent biocompatibility and ROS scavenging ability. A recent article indicated that a hydrogel incorporated with V_2_C nanosheets could protect articular cartilage by providing a lubrication effect and scavenging excess ROS [27]. Therefore, MXene is a promising biomaterial for OA management.

In this study, we characterized Ti_3_C_2_T_x_ nanosheets and explored their potential in treating OA. Ti_3_C_2_T_x_ nanosheets demonstrated excellent antioxidant capacity and contributed to balancing anabolic activities and catabolic activities in chondrocytes. Finally, intra-articular injection of Ti_3_C_2_T_x_ nanosheets effectively ameliorated OA progression. Hence, we believed that Ti_3_C_2_T_x_ nanosheets could be a promising option for OA treatment.

## 2. Materials and Methods

### 2.1. Cell Isolation and Culture

Chondrocytes were isolated from 5-day-old rats. Briefly, Sprague Dawley (SD) rats were humanely euthanized using the cervical dislocation method. Then, under sterile conditions, microsurgical instruments were used to sequentially dissect the knee joint area of the rat’s hind leg to expose the tibial plateau cartilage. The cartilage was carefully separated to ensure that the isolated cartilage was uniform, transparent, and elastic, avoiding contamination with other tissues. The isolated cartilage pieces were washed 3 times. Then, 0.25% trypsin was used to digest the cartilage tissue for 30 min, and the cartilage was subsequently digested by 0.2% collagenase II in a 37 °C incubator for 4–6 h. After digestion, the supernatant was collected and centrifuged (1500 rpm, 5 min). The chondrocytes were resuspended in DMEM supplemented with 10% FBS (Hyclone, Logan, UT, USA) and 1% penicillin-streptomycin (Boster Bio, Wuhan, China) to obtain a chondrocyte suspension. The chondrocytes were cultured in a humidified incubator (Thermo, Waltham, MA, USA) maintained at 37 °C with 5% CO_2_. Passages were conducted once the cells reached 80–90% confluency.

### 2.2. Ti_3_C_2_T_x_ Nanosheets Characterization

Ti_3_C_2_T_x_ nanosheets were commercially purchased from XFNANO company, Nanjing, China. The morphology of Ti_3_C_2_T_x_ nanosheets was first observed using Transmission Electron Microscopy (TEM; FEI Talos F200X G2, Hillsboro, OR, USA). Dynamic Light Scattering Analyzer (DLS; Malvern Zetasizer Nano ZS90, Malvern, UK) was used to analyze the diameter distribution and Zeta potential of Ti_3_C_2_T_x_ nanosheets. The thickness of Ti_3_C_2_T_x_ nanosheets was determined using an Atomic Force Microscope (AFM; Bruker Dimension ICON, Billerica, MA, USA).

### 2.3. Cell Viability Test

The viability and proliferation levels of the chondrocytes were measured using the Cell Counting Kit-8 (CCK-8) (Solarbio, Beijing, China). In brief, after incubating with Ti_3_C_2_T_x_ nanosheets for 24 h, the cell culture medium was discarded and 90 μL of fresh DEME/F12 complete medium along with 10 μL of CCK-8 reagent were added. After 60 min, the absorbance of the medium was determined using a microplate reader at 450 nm.

### 2.4. Lived/Dead Cell Staining

Chondrocytes were first seeded into 96-well cell culture plates. After applying different interventions to the chondrocytes for 24 h, the chondrocytes were stained using the Calcein/PI Cell Viability Assay Kit (Beyotime, Shanghai, China). Briefly, the culture medium was removed at the predetermined time point and washed three times using PBS (Arlington, VA, USA). Then, 200 μL of staining solution was added to each well. The plates were incubated in a cell incubator away from light for 20 min. The live/dead status of the chondrocytes was imaged using a fluorescence microscope.

### 2.5. Antioxidant Efficiency of Ti_3_C_2_T_x_ Nanosheets

The ROS scavenging efficiency of Ti_3_C_2_T_x_ nanosheets was first assessed using the Total Antioxidant Capacity Kit (Beyotime, China). In brief, 10 μL Ti_3_C_2_T_x_ nanosheets were added to ABTS working solution and incubated for 5 min away from light. Subsequently, the absorbance (414 nm) was detected using a microplate reader. The intracellular ROS level in chondrocytes was determined using a cell-permeable probe DCFH-DA (MedChemExpress, Junction, NJ, USA). After incubating the chondrocytes with or without H_2_O_2_ and/or Ti_3_C_2_T_x_ nanosheets for 24 h, the chondrocytes were washed three times. Then, the chondrocytes were stained in the dark with 10 μM DCFH-DA for 30 min and Hoechst 33342 (MedChemExpress, USA) for 10 min. Finally, the stained chondrocytes were observed and imaged using the fluorescence microscope.

### 2.6. Quantitative Real-Time Polymerase Chain Reaction

The total RNA of chondrocytes was extracted using the Total RNA Kit (Omega Bio-tek, Norcross, GA, USA). The concentration and purity of the extracted chondrocyte RNA were measured using a NanoDrop 1000 reader (Thermo Scientific, Waltham, MA, USA). Next, cDNA was synthesized using the ReverTra Ace^®^ qPCR RT Master Mix (TOYOBO, Tokyo, Japan). The cDNA of all detected genes was quantified using the Quantagene q225 Real-Time PCR system. Each experiment was performed in triplicate to obtain average data. The relative mRNA expression levels were calculated using the ΔΔCT method.

### 2.7. Immunofluorescence Staining

The chondrocytes were gently washed three times and fixed using 4% paraformaldehyde for 15 min. Then, the chondrocytes were subsequently treated with an immunostaining permeabilization buffer with Triton X-100 (Beyotime, China) for 10 min. Next, the chondrocytes were blocked by 2% bovine serum albumin for 30 min. Then, the chondrocytes were incubated with rabbit anti-Col2/MMP13 primary antibody at 4 °C overnight, followed by a 1 h incubation with Cy3-conjugated or FITC-conjugated goat anti-rabbit immunoglobulin G (Thermo Scientific, USA). DAPI (Boster, China) working solution was employed to stain the nuclei. Finally, the chondrocytes were observed and imaged by a fluorescence microscope.

### 2.8. RNA Sequencing

The total RNA from the chondrocytes was extracted using the TRIzol reagent, and the sample quality and integrity were subsequently tested. Then, an RNA sequencing library was established after the quantitative analysis of RNA samples. Igenebook Company (Wuhan, China) helped with processing the data. The heatmap, volcano plot, and dot plot of the Gene Ontology (GO) enrichment analysis and Kyoto Encyclopedia of Genes and Genomes (KEGG) pathway enrichment analysis were plotted by https://www.bioinformatics.com.cn.

### 2.9. Western Blotting

After receiving various treatments, the total protein of the chondrocytes was extracted using a radioimmunoprecipitation assay (RIPA) buffer (Boster Bio, China) containing protease inhibitors, phosphatase inhibitors, and EDTA. A BCA protein quantitative kit (Beyotime, China) was applied to determine the protein concentration. Through SDS-PAGE gel electrophoresis, the proteins were subsequently separated and transferred from the gel to a 0.22 μm polyvinylidene difluoride membrane (Thermo Scientific, USA) in a transfer buffer. Then, the membrane containing transferred protein was blocked using TBST containing 5% nonfat milk for 1 h. After blocking, the membrane was incubated with primary antibodies at 4 °C overnight. The primary antibodies used include Anti-PI3 Kinase p85antibody (Proteintech, Wuhan, China), Anti-Phospho-PI3 Kinase p85 (Abcam, Cambridge, UK), Anti Akt antibody (Abcam, UK), Anti Phospho-Akt antibody (Abcam, UK), Anti-GAPDH antibody (Proteintech, China). Then, the membrane was washed 3 times by TBST and was subsequently incubated with HRP-conjugated IgG (1:2000; Thermo Scientific, USA) for 1 h at room temperature. Finally, the membrane was further washed 3 times using TBST and was visualized using ECL chemiluminescence assay kits (Thermo Scientific, USA).

### 2.10. Induction of Rat OA Model

The animal experiment was approved by the Animal Research Committee of Huazhong University of Science and Technology. Eighteen Sprague-Dawley (SD) rats were randomly divided into three groups (DMM, Ti_3_C_2_T_x_+, and Ti_3_C_2_T_x_++). All rats received surgical destabilization of the medial meniscus (DMM). In brief, the rats were anesthetized with 3% pentobarbital sodium (40 mg/kg) and subjected to medial meniscectomy of the right knee joint. Two weeks post-surgery, the right joint cavities of rats in the DMM group, Ti_3_C_2_T_x_+ group, and Ti_3_C_2_T_x_++ group were injected with PBS, Ti_3_C_2_T_x_ nanosheets (10 μg/mL), and Ti_3_C_2_T_x_ nanosheets (20 μg/mL), respectively. Additional injections were administered once every 2 days for 2 weeks. All rats were humanely euthanized eight weeks after surgery. The right knee joints of all rats were collected and fixed with a 4% paraformaldehyde solution for further experiments.

### 2.11. Statistical Analysis

All data were analyzed by GraphPad Prism 9 and Origin 2023 software and were shown in the form of the mean ± standard deviation (SD). All experiments were conducted with at least three replicates. For the significance analyses, a one-way variance (ANOVA) test was performed to compare multiple groups. * *p* < 0.05, ** *p* < 0.01, *** *p* < 0.001 were considered statistically significant, and ns. indicates no significance.

## 3. Results

### 3.1. Characterization of Ti_3_C_2_T_x_ Nanosheets

2D Ti_3_C_2_T_x_ nanosheets were first characterized by TEM. The TEM image indicated that Ti_3_C_2_T_x_ displayed a typical 2D layered structure (Figure 1A). In addition, Ti_3_C_2_T_x_ nanosheets had a mean lateral dimension of 521.43 nm (n = 10), whereas the DLS results demonstrated that Ti_3_C_2_T_x_ nanosheets had a mean size of 338.2 nm (Figure 1B). This may be because the ultrasonic treatment prior to DLS analysis further decreased the size of Ti_3_C_2_T_x_ nanosheets. Further, DLS analysis indicated that Ti_3_C_2_T_x_ nanosheets had a peak zeta potential of −28.7 mV (Figure 1C). Next, the thickness of Ti_3_C_2_T_x_ nanosheets was evaluated by AFM. As demonstrated in Figure 1D, the ultrathin Ti_3_C_2_T_x_ nanosheet had a thickness of 2.91 nm. The AFM analysis revealed the monolayered or few-layered structure of Ti_3_C_2_T_x_ nanosheets. The C, O, and Ti elements of the Ti_3_C_2_T_x_ nanosheets were characterized by HAADF-STEM images and elemental mappings (Figure 1E).

### 3.2. Cytocompatibility and ROS Scavenging Ability of Ti_3_C_2_T_x_ Nanosheets

The cytocompatibility was first evaluated through the CCK-8 assay. According to Figure 2, the proliferation of chondrocytes was slightly inhibited by Ti_3_C_2_T_x_ nanosheets when the concentration was 40 μg/mL. However, Ti_3_C_2_T_x_ nanosheets had no impact on the chondrocyte proliferation when the concentration was under 20 μg/mL. Further, the live/dead cell staining demonstrated that Ti_3_C_2_T_x_ nanosheets did not induce chondrocyte death even when the concentration was up to 40 μg/mL (Figure 3). According to the results of the CCK-8 assay and live/dead cell staining, the treatment concentration of Ti_3_C_2_T_x_ nanosheets was determined (10 μg/mL and 20 μg/mL) for subsequent experiments.

The ROS scavenging ability of the Ti_3_C_2_T_x_ nanosheets was evaluated using 2,2′-azino-bis(3-ethylbenzthiazoline-6-sulfonic acid (ABTS) and 2,2-Diphenyl-1-picrylhydrazyl (DPPH). Ti_3_C_2_T_x_ nanosheets showed excellent ROS scavenging ability, and the scavenging efficiency increased with the concentration of Ti_3_C_2_T_x_ nanosheets becoming elevated (Figure 4A,B). Trolox, which is a vitamin E analog with strong antioxidant properties, served as the positive control in this study. Further, the intracellular ROS scavenging ability of Ti_3_C_2_T_x_ nanosheets was determined using 2,7-Dichlorodihydrofluorescein diacetate (DCFH-DA). As illustrated in Figure 4C,E, the elevated intracellular ROS level in the chondrocytes was inhibited by Ti_3_C_2_T_x_ nanosheets. As chondrocytes exposed to excess ROS would inevitably show cell apoptosis, we determined whether Ti_3_C_2_T_x_ nanosheets could protect chondrocytes from oxidative stress and inhibit the chondrocyte apoptosis elicited by excessive ROS exposure. The results of live/dead cell staining demonstrated that exposure to hydrogen peroxide (H_2_O_2_, 100 μM) would inevitably induce cell death, while the pro-apoptosis effect of H_2_O_2_ was significantly inhibited by Ti_3_C_2_T_x_ nanosheets (Figure 4D,F).

### 3.3. Ti_3_C_2_T_x_ Nanosheets Restored the Cellular Homeostasis and Inhibited Inflammation

The effect of Ti_3_C_2_T_x_ nanosheets on IL-1β-challenged chondrocytes was first evaluated by qRT-PCR. The gene expression levels of aggrecan (ACAN), collagen II (Col2), matrix metalloproteinase 13 (MMP13), inducible nitric oxide synthase (iNOS), cyclooxygenase-2 (COX-2) were evaluated. ACAN and Col2 are major matrix components in cartilage and serve as important biomarkers for chondrocyte anabolism. In contrast, MMP13 is one of the important matrix metalloproteinases in OA progression and serves as a biomarker for chondrocyte catabolism. iNOS and COX-2 are well-known biomarkers of inflammation response. According to the qRT-PCR results, it could be seen that the Ti_3_C_2_T_x_ nanosheets significantly promoted the gene expression of ACAN and Col2 while downregulating the gene expression level of MMP13 (Figure 5A,B,E). Importantly, Ti_3_C_2_T_x_ nanosheets also prominently inhibited the inflammation elicited by IL-1β (Figure 5C,D). The alternation induced by the Ti_3_C_2_T_x_ nanosheets in protein expression level was determined by immunofluorescence. As illustrated in Figure 5F, the downregulated trend in Col2 expression was reversed by the Ti_3_C_2_T_x_ nanosheets as evidenced by the enhanced immunofluorescence intensity by Ti_3_C_2_T_x_ nanosheets. Meanwhile, Ti_3_C_2_T_x_ nanosheets significantly inhibited the MMP13 protein expression levels elevated by IL-1β (Figure 5G). Consistent with the qRT-PCR result, Ti_3_C_2_T_x_ nanosheets prominently reduced the iNOS expression in chondrocytes (Figure 5H).

### 3.4. RNA Sequencing Analysis of Ti_3_C_2_T_x_ Nanosheets against IL-1β-Challenged Chondrocytes

To explore the potential mechanism underlying Ti_3_C_2_T_x_ nanosheet-mediated therapeutic effect, we performed RNA sequencing on IL-1β-challenged OA chondrocytes treated with or without Ti_3_C_2_T_x_ nanosheets. According to the volcano plot and heat map analyses, a total of 1587 differently expressed genes (DEGs) emerged, among which 482 genes were up-regulated, and 1105 genes were down-regulated after treatment with Ti_3_C_2_T_x_ nanosheets (Figure 6A,B). Gene Ontology (GO) enrichment analysis were conducted to explore which cellular processes, functions, and components were altered by the Ti_3_C_2_T_x_ nanosheets. GO terms probably related to cartilage homeostasis were chosen and shown by three GO categories including biological processes (BP), cellular components (CC), and molecular functions (MF). The extracellular region part and extracellular space were significantly enriched after treatment with Ti_3_C_2_T_x_ nanosheets (Figure 6C). In addition, inflammation response and regulation of inflammation response were enriched by Ti_3_C_2_T_x_ nanosheets (Figure 6C). The GO enrichment analysis was consistent with the aforementioned results demonstrating that the Ti_3_C_2_T_x_ nanosheets significantly promoted cartilage matrix synthesis while inhibiting cellular inflammation. Next, the Kyoto Encyclopedia of Genes and Genomes (KEGG) pathway enrichment analysis was performed to explore the Ti_3_C_2_T_x_ nanosheets-mediated cellular pathways. As shown in Figure 6D, Ti_3_C_2_T_x_ nanosheet treatment induced marked activation of phosphatidylinositol 3-kinase/protein kinase B (PI3K)-Akt pathways. And the western blotting results also confirmed the activation of PI3K-Akt pathways by Ti_3_C_2_T_x_ nanosheets in chondrocytes (Appendix A).

### 3.5. Ti_3_C_2_T_x_ Nanosheets Effectively Ameliorated OA Progression In Vivo

A rat OA model was established to determine the in vivo effect of Ti_3_C_2_T_x_ nanosheets. A total of 18 rats were subjected to surgical destabilization of the medial meniscus (DMM) and were randomly allocated into 3 groups (DMM, Ti_3_C_2_T_x_+ (10 μg/mL), Ti_3_C_2_T_x_++ (20 μg/mL) Ti_3_C_2_T_x_). According to hematoxylin-eosin (H&E) staining, the cartilage in the DMM group showed severe destruction and deformation, while Ti_3_C_2_T_x_ treatment significantly inhibited the cartilage degeneration induced by DMM surgery (Figure 7A). In addition, the results of safranine O-fast green staining and toluidine blue staining verified the prominent effect of Ti_3_C_2_T_x_ treatment on inhibiting cartilage loss (Figure 7A). The in vivo therapeutic effect of Ti_3_C_2_T_x_ was also supported by the Osteoarthritis Research Society International (OARSI) score. The OARSI score in the DMM group was reduced by 44.1% in comparison to the Ti_3_C_2_T_x_+ group and was reduced by 52% in comparison to the Ti_3_C_2_T_x_++ group (Figure 7B). The immunohistochemistry results also demonstrated that Ti_3_C_2_T_x_ could promote the Col2 expression while inhibiting the MMP13 expression (Figure 7C–F). In addition, it could be seen that there was no difference between the therapeutic effect of Ti_3_C_2_T_x_+ (10 μg/mL) and Ti_3_C_2_T_x_++ (20 μg/mL). Conclusively, Ti_3_C_2_T_x_ treatment produced an in vivo therapeutic effect on OA progression induced by DMM.

## 4. Discussion

Recently, 2D MXene nanosheets have demonstrated great potential in treating various diseases and emerged as a promising biomaterial for biomedical applications [28]. MXene nanosheets can act as enzymes to alleviate cellular oxidative stress. By mimicking natural enzymes like superoxide dismutase (SOD), catalase (CAT), peroxidase (POD), and glutathione peroxidase (GPx), MXene nanosheets could effectively scavenge overproduced ROS elicited by pathological cues [20,29]. Therefore, MXene nanosheets have a therapeutic effect on ROS-driven inflammation and ROS-induced degenerative changes [20]. Excess ROS level also contributes to OA development, and biomaterials capable of scavenging ROS show the potential to delay OA progression [30,31].

In this study, we characterized Ti_3_C_2_T_x_ nanosheets and their potential to treat OA. Ti_3_C_2_T_x_ nanosheets demonstrated acceptable biocompatibility. Though a concentration of 40 μg/mL would lead to slight inhibition in chondrocyte proliferation, a concentration below 20 μg/mL had no detrimental effect on chondrocyte proliferation. Notably, another study reported that polyvinylpyrrolidone (PVP)-modified Ti_3_C_2_T_x_ nanosheets posed no impact on cell proliferation even with a concentration of up to 80 μg/mL [21]. This may be due to improved biocompatibility and physiological stability resulting from PVP modification. Furthermore, Ti_3_C_2_T_x_ nanosheets below 40 μg/mL did not induce cell death, which indicated the biosafety of Ti_3_C_2_T_x_ nanosheets.

In vitro experiments verified that Ti_3_C_2_T_x_ nanosheets could protect chondrocytes from oxidative stress and degeneration. Firstly, Ti_3_C_2_T_x_ nanosheets demonstrated efficient scavenging ability toward ABTS and DPPH, two indicators widely applied for the determination of antioxidant activities. Moreover, Ti_3_C_2_T_x_ nanosheets with antioxidant capacity could downregulate the intracellular ROS level thus reducing cell death resulting from oxidative stress. A proper balance between catabolism and anabolism is vital for healthy cartilage [32]. However, the homeostasis of catabolism and anabolism is disturbed in OA patients, which eventually leads to cartilage loss and degeneration [32]. This study showed that Ti_3_C_2_T_x_ nanosheets contributed to restoring cartilage homeostasis by promoting anabolism while inhibiting catabolism in chondrocytes. Ti_3_C_2_T_x_ nanosheets promoted impaired anabolic activities in IL-1β-induced OA chondrocytes while inhibiting enhanced catabolic activities. Ti_3_C_2_T_x_ nanosheets protect cartilage from degeneration by inhibiting cellular inflammation. Low-grade inflammation is a crucial mediator of the pathogenesis and pain associated with OA [33,34]. It is believed that the prominent anti-inflammation capacity of Ti_3_C_2_T_x_ nanosheets is tightly associated with their ROS scavenging ability as overproduced ROS drives elevated production of proinflammatory cytokine [35].

PI3K/AKT is an important signaling pathway that is involved in various essential cellular processes, such as cell cycle, inflammation, metabolism, and apoptosis [36,37,38]. The PI3K/AKT signaling pathway also contributes to cartilage homeostasis, and the presence of a downregulated PI3K/AKT pathway was verified in human OA cartilage [38,39]. Further studies have demonstrated that the activation of the PI3K/AKT pathway was conducive to restoring balance between anabolic activities and catabolic activities in cartilage. For example, increased phosphorylation of Akt resulting from PTEN inhibition significantly promoted proteoglycan synthesis under oxidative stress [40]. In addition, activation of the PI3K/AKT pathway mediated the chondroprotection of Mg^2+^ on IL-1β challenged chondrocytes [41]. Similarly, we first identified that Ti_3_C_2_T_x_ nanosheets led to marked activation of the PI3K/AKT pathway in chondrocytes by RNA sequencing and KEGG pathway enrichment analysis. In addition, elevated phosphorylation of PI3K and AKT induced by Ti_3_C_2_T_x_ nanosheets was identified by Western blotting. Finally, Ti_3_C_2_T_x_ nanosheets were delivered through direct injection rather than delivery by hydrogel. This is because this study aims to explore the therapeutic potential of Ti_3_C_2_T_x_ nanosheets, while the hydrogel may provide boundary lubrication for mitigating OA progression. In addition, Ti_3_C_2_T_x_ nanosheets easily oxidize when they are kept under wet conditions. Therefore, increased oxidation of Ti_3_C_2_T_x_ nanosheets might result from the controlled release property of the hydrogel system. The oxidation of Ti_3_C_2_T_x_ nanosheets significantly increases their cytotoxicity [42]. Therefore, the Ti_3_C_2_T_x_ nanosheets were directly injected to verify their in vivo benefits. Through histological staining and immunohistochemistry, we found that Ti_3_C_2_T_x_ nanosheets could inhibit the loss of glycosaminoglycan and proteoglycan content and the promotion of catabolic activities (MMP13), eventually preventing cartilage from degeneration.

According to our results and the study by Zhang et al., both Ti_3_C_2_T_x_ nanosheets and V_2_CT_x_ nanosheets demonstrated therapeutic potential in treating OA. In their study, V_2_CT_x_ nanosheets acted as an antioxidant and provided boundary lubrication in a hybrid hydrogel system, and the V_2_CT_x_ nanosheets prevented the hydrogel-protected cartilage from degeneration by activating the Nrf2-ARE signaling pathway [27]. While our study indicated that simple Ti_3_C_2_T_x_ nanosheets are able to mitigate OA progression by scavenging overproduced ROS. No experiment was conducted to demonstrate that Ti_3_C_2_T_x_ nanosheets or V_2_CT_x_ nanosheets could be taken up by chondrocytes. However, studies have reported that both Ti_3_C_2_T_x_ nanosheets and V_2_CT_x_ nanosheets could be taken up by other cell types like HepG2 cells and macrophage [14,43]. Hence, we believed that both Ti_3_C_2_T_x_ nanosheets and V_2_CT_x_ nanosheets could be engulfed by chondrocytes thus triggering the following cellular pathway. Various MXene such as Ti_3_C_2_T_x_, V_2_CT_x_, and Nb_2_CT_x_ have demonstrated promising results in the field of biomedicine. Further studies are needed to clarify the best MXene type for OA treatment.

Though MXene has shown great potential in treating various diseases, the translation of MXene into biomedical applications is still in its infancy [44]. The main reason behind this is that the number of studies on the biocompatibility and cytocompatibility of MXene is very limited [44]. Our results indicated that Ti_3_C_2_T_x_ nanosheets posed no damage to chondrocytes, while the potential in vivo effect of Ti_3_C_2_T_x_ nanosheets on excretory organs is not well studied. Noteworthily, a previous study demonstrated that intravenously injected Ti_3_C_2_T_x_ nanosheets could be degraded and excreted through the urine and feces [21]. Also, they further indicated that PVP-modified Ti_3_C_2_T_x_ nanosheets had no impact on mice organs like the heart, liver, spleen, lung, and kidneys [21]. But long-term studies and large animal models are still needed to assess their biosafety, which would accelerate the clinical translation of MXene in OA treatment. In addition, a major concern about the application of MXene is its easy oxidation and instability. A recent study using Raman spectroscopy showed Ti_3_C_2_T_x_ oxidation during its degradation [15]. While the oxidation of MXene is believed to be harmful to cells and impairs cell viability, surface modification with polymers or polydopamine has been reported to stabilize MXene and decrease its cytotoxicity [42,44,45]. Therefore, future studies exploring MXene with surface modifications suitable for articular cartilage would further contribute to the application of MXene in OA treatment.

## 5. Conclusions

In this study, we identified the antioxidant capacity of Ti_3_C_2_T_x_ nanosheets and their therapeutic potential in ameliorating OA progression. Ti_3_C_2_T_x_ nanosheets could effectively scavenge overproduced ROS and protect chondrocytes from oxidative stress-induced cell death. Ti_3_C_2_T_x_ nanosheets also contributed to restoring the homeostasis of anabolism and catabolism in chondrocytes. Furthermore, we demonstrated the potential mechanism underlying Ti_3_C_2_T_x_ nanosheet-mediated chondroprotective effects. Finally, Ti_3_C_2_T_x_ nanosheets effectively protect cartilage from degeneration in vivo. Conclusively, the in vitro and in vivo experiments supported that Ti_3_C_2_T_x_ nanosheets could be a promising biomaterial for OA treatment.

## Figures and Tables

**Figure 1 nanomaterials-14-01572-f001:**
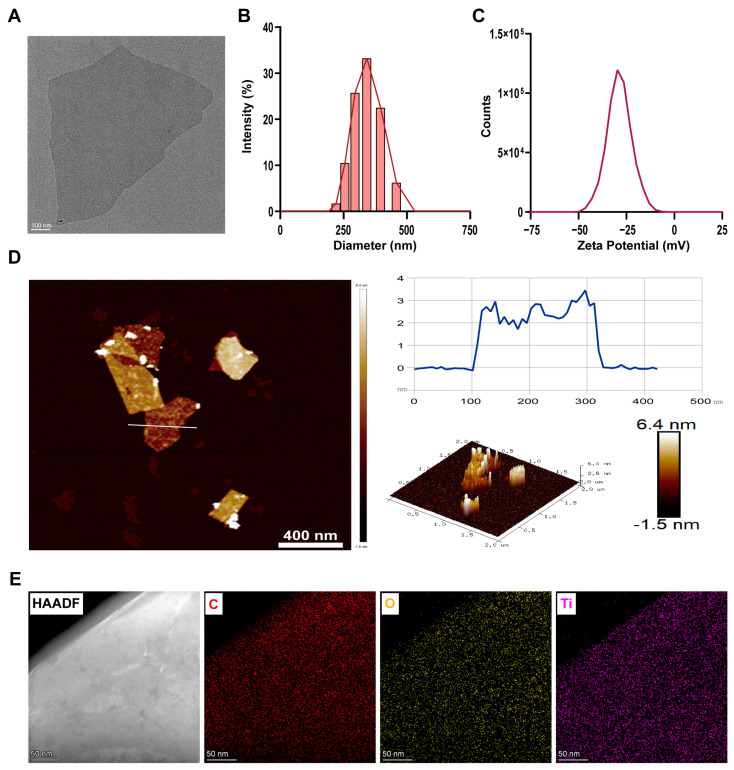
Characterization of Ti_3_C_2_T_x_ nanosheets. (**A**): TEM image of Ti_3_C_2_T_x_ nanosheets. (**B**): Size distribution of Ti_3_C_2_T_x_ nanosheets according to DLS analysis. (**C**): Zeta potential of Ti_3_C_2_T_x_ nanosheets. (**D**): Representative AFM image of Ti_3_C_2_T_x_ nanosheets. (**E**): Energy-dispersive X-ray spectroscopy elemental mapping of Ti_3_C_2_T_x_ nanosheets.

**Figure 2 nanomaterials-14-01572-f002:**
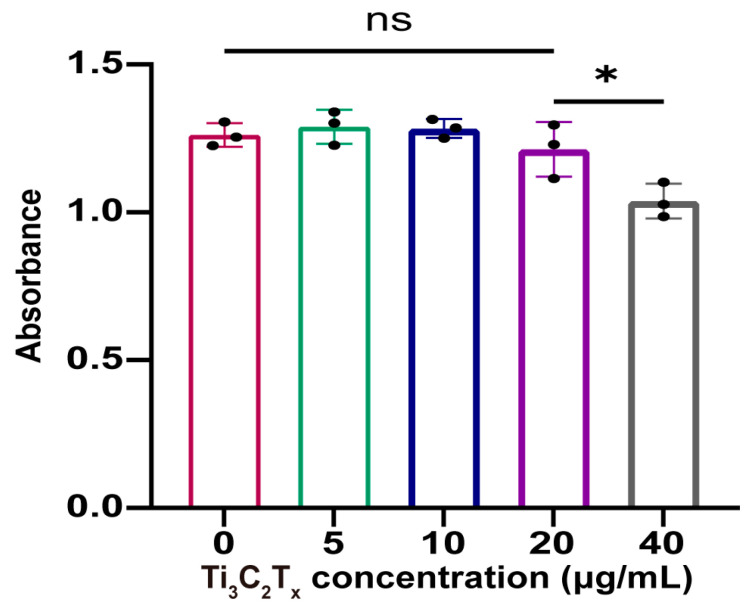
CCK-8 analysis to evaluate chondrocyte viability and proliferation after Ti_3_C_2_T_x_ nanosheets treatment. n = 3. Data are presented as the mean ± SD. (* *p* < 0.05; ns: no significance).

**Figure 3 nanomaterials-14-01572-f003:**
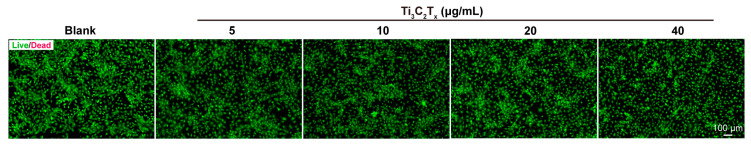
The live/dead staining of rat chondrocytes after Ti_3_C_2_T_x_ nanosheet treatment.

**Figure 4 nanomaterials-14-01572-f004:**
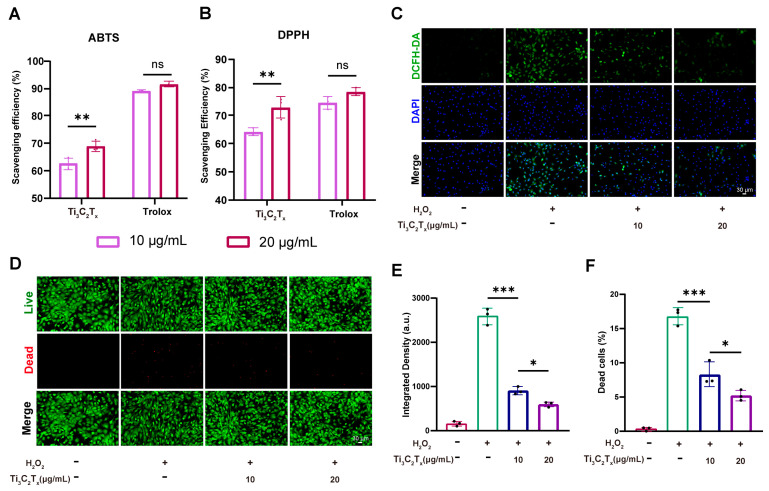
Ti_3_C_2_T_x_ nanosheets exhibited ROS scavenging ability and protected chondrocytes from cell death. (**A**): ABTS scavenging efficiency of Ti_3_C_2_T_x_ nanosheets and Trolox. (**B**): DPPH scavenging efficiency of Ti_3_C_2_T_x_ nanosheets and Trolox. (**C**): Intracellular ROS level in chondrocytes with and without Ti_3_C_2_T_x_ nanosheet treatment. (**D**): The live/dead staining of rat chondrocytes with and without Ti_3_C_2_T_x_ nanosheet treatment. (**E**): Quantitative analysis of the intracellular ROS level. (**F**): Quantitative analysis of the live/dead staining. n = 3. Data are presented as the mean ± SD. (* *p* < 0.05; ** *p* < 0.01; *** *p* < 0.001; ns: no significance).

**Figure 5 nanomaterials-14-01572-f005:**
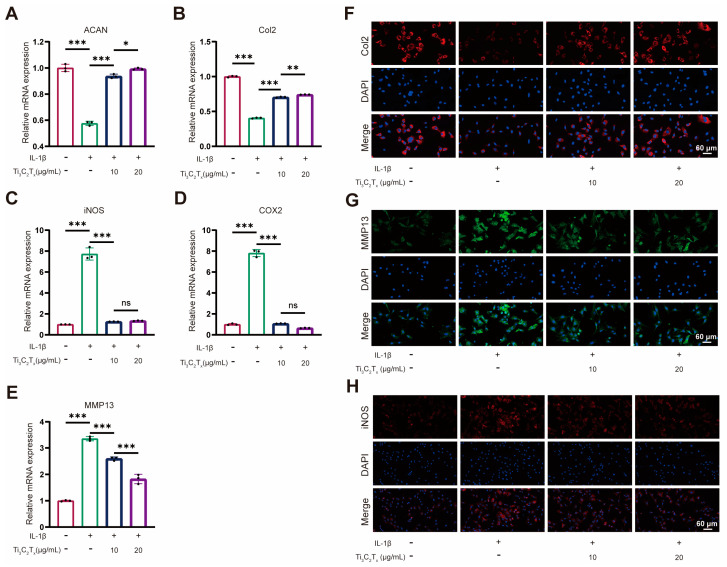
Ti_3_C_2_T_x_ nanosheets restored matrix homeostasis and inhibited inflammation in chondrocytes. (**A**–**E**): Relative mRNA expression levels of ACAN, Col2, iNOS, COX2, and MMP13. (**F**): Immunofluorescent staining of Col2 in chondrocytes. (**G**): Immunofluorescent staining of MMP13 in chondrocytes. (**H**): Immunofluorescent staining of iNOS in chondrocytes. n = 3. Data are presented as the mean ± SD (* *p* < 0.05; ** *p* < 0.01; *** *p* < 0.001; ns: no significance).

**Figure 6 nanomaterials-14-01572-f006:**
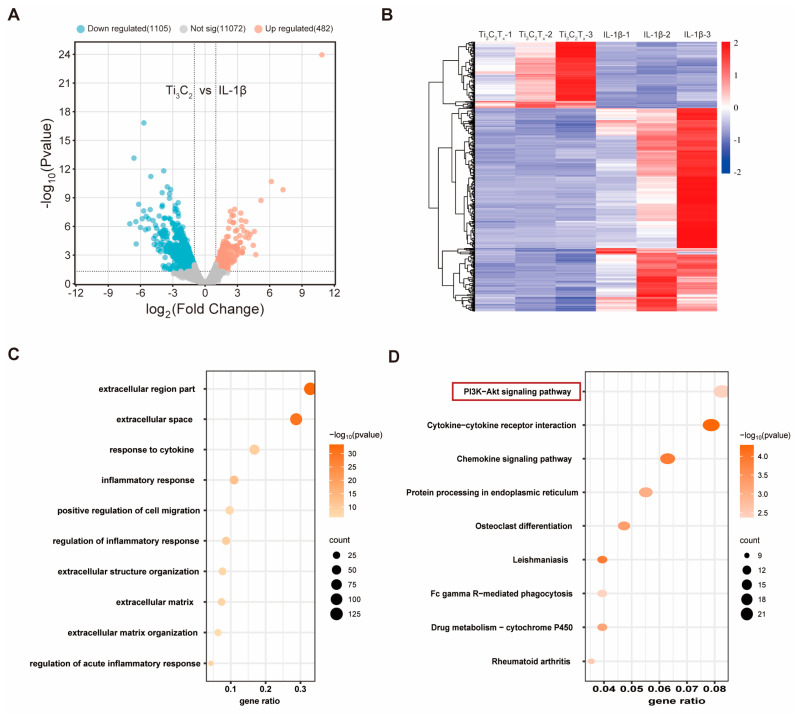
Potential mechanism underlying the therapeutic effect of Ti_3_C_2_T_x_ nanosheets on chondrocytes. (**A**): Volcano plot displayed differentially expressed genes (DEGs). *p* < 0.05, |fold change| ≥ 1.5. (**B**): Heatmap of DEGs between Ti_3_C_2_T_x_ nanosheets + IL-1β and IL-1β. (**C**): GO pathways associated with cartilage homeostasis. (**D**): KEGG enrichment analysis of Ti_3_C_2_T_x_ nanosheets + IL-1β vs. IL-1β.

**Figure 7 nanomaterials-14-01572-f007:**
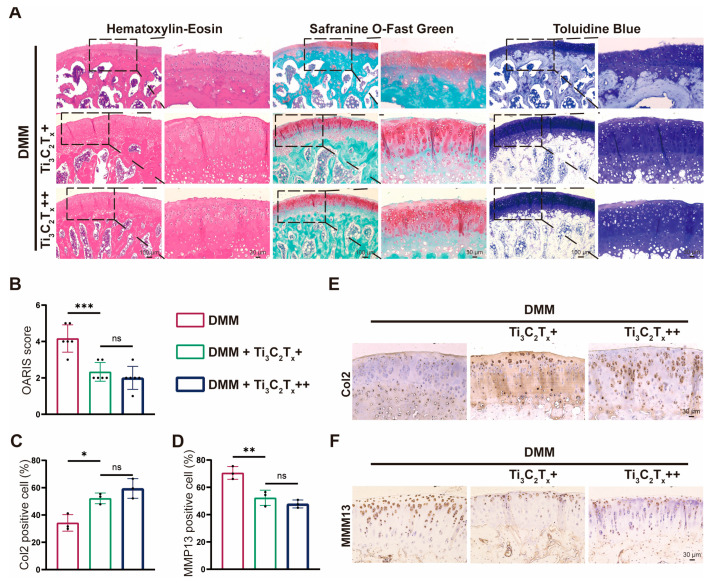
Ti_3_C_2_T_x_ nanosheets effectively mitigated OA progression in vivo. (**A**): Representative images of H&E staining, safranin O-fast green staining, and toluidine blue staining. The two images presented for each staining in a group are of the same sample at different magnifications. (**B**): OARIS score of articular cartilage from each group (n = 6). (**C**): Quantitative analysis of the Col2 positive cell (n = 3). (**D**): Quantitative analysis of the MMP13 positive cell (n = 3). (**E**): Immunohistochemical staining of Col2 protein in articular cartilage. (**F**): Immunohistochemical staining of MMP13 protein in articular cartilage. Data are presented as the mean ± SD (* *p* < 0.05; ** *p* < 0.01; *** *p* < 0.001; ns: no significance).

## Data Availability

The data that support the findings of this study are available from the corresponding author upon reasonable request.

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
