# Peer review of "2D MXene Nanosheets with ROS Scavenging Ability Effectively Delay Osteoarthritis Progression"

_nanomaterials, 2024, doi:10.3390/nano14191572_

Round 1

Reviewer 1 Report

Comments and Suggestions for Authors

The research article: 2D Ti3C2 nanosheets with ROS scavenging ability effectively delay osteoarthritis progression

This research focuses on titanium-based MXene nanosheets' therapeutic potential in delaying osteoarthritis (OA) progression. In vitro experiments indicate the strong ROS scavenging capacity of Ti3C2 nanosheets and their acceptable biocompatibility. In addition, Ti3C2 nanosheets demonstrate a prominent anti-inflammatory effect and the ability to restore homeostasis between anabolic activities and catabolic activities in chondrocytes. Furthermore, RNA sequencing reveals the potential mechanism underlying the Ti3C2-nanosheets-mediated therapeutic effect. Finally, a rat OA model verifies the in vivo curative effect of Ti3C2 nanosheets. Histological staining and immunohistochemical analyses indicate that Ti3C2 nanosheets effectively ameliorate OA progression. The experiments that were conducted were comprehensive and well-explained. However, there are some remarks about this work:

1)     It would be more accurate to use the Ti3C2Tx formula instead of Ti3C2, where Tx marks an unknown functional group amount, which is meaningful.

2)     In the abstract and title, MXenes should be mentioned.

3)     In this study, Ti3C2Tx is investigated as an antioxidant; however, it is well known that MXenes are unstable and easily oxidizing. It would be worth mentioning more properties and highlighting the stability issue of MXenes in the introduction. The recent article about MXenes stability investigation by Raman spectroscopy could give a useful insights for the Authors: https://doi.org/10.1021/acsnano.4c02150

4)     It would be worth extending the introduction part and explaining MXenes properties and oxidation.

Author Response

Dear reviewer

On behalf of all the contributing authors, I would like to express our sincere appreciations of reviewers’ constructive comments concerning our manuscript. These comments are all valuable and helpful for improving our article. According to the reviewers’ comments, we have made extensive modifications to our manuscript and supplemented data to make our results convincing. In the revised version, changes to our manuscript were all highlighted within the document by using red-colored text and an underscore. Point-by-point responses to two nice reviewers are listed below this letter.

Reviewer 1

1)It would be more accurate to use the Ti3C2Tx formula instead of Ti3C2, where Tx marks an unknown functional group amount, which is meaningful.

Response: Thank you for your correction of the inappropriate formula. All the Ti3C2 formula have been replaced by Ti3C2Tx.

2)In the abstract and title, MXenes should be mentioned.

Response: Thank you for your comment. MXenes were mentioned in the abstract and title. (Line 12-line 15)

3)In this study, Ti3C2Tx is investigated as an antioxidant; however, it is well known that MXenes are unstable and easily oxidizing. It would be worth mentioning more properties and highlighting the stability issue of MXenes in the introduction. The recent article about MXenes stability investigation by Raman spectroscopy could give a useful insights for the Authors: https://doi.org/10.1021/acsnano.4c02150

Response: Thank you for your valuable comment. The recent study (https://doi.org/10.1021/acsnano.4c02150) provides a useful insight for us and deepen our knowledge about Ti3C2Tx. Given that the stability issue and oxidation of MXenes is tightly related to the its cytotoxicity, we highlight the oxidation of MXenes in the discussion of the manuscript, and cite the given article to support our discussion. (Line 407-line 411)

4) It would be worth extending the introduction part and explaining MXenes properties and oxidation.

Response: Thank you for your valuable comment. We extend the introduction part to introduce more properties of MXenes, and extend the discussion part to discuss the oxidation issue of MXenes. (Line 49-line 52; line 407-line 411)

Reviewer 2 Report

Comments and Suggestions for Authors

The authors present data to indicate that, like other types of nanosheets, Ti3C2 nanosheets have potential to mediate harmful ROS in the body. They further demonstrate this with in vitro data and an in vivo model of osteoarthritis. Several points need to be addressed and further discussed:

1) In the measurement of antioxidant efficiency, it would be helpful to have a positive control tested alongside the nanosheets, to give a comparator that is relatable to general literature.

2) The results from the osteoarthritis model require greater explanation, particularly as the negative control group (DMM) did not appear to have significant loss of cartilage structure after 8 weeks in Figure 7A. Another panel showing what undamaged, healthy cartilage looks like in the rat should be added to the figure. Also, please amend Figure 7 caption to make it clear that the two images presented for each group/staining in panel A are the same sample at different magnifications.

3) In the discussion on the mechanism of action for the nanosheets, western blotting of PI3K and AKT was mentioned, but it was not presented in the methods or data. This must be corrected, preferably with the additional data.

4) The authors present evidence from literature for the potential benefits of nanosheets. However, there is no discussion on the reason why the authors believe Ti3C2 would be better than V2C, or a comparison to the results of the previous study using V2C. Are both of these nanosheet types believed to act via the same cellular pathway? Are they believed to be taken up by cells, or to act at the cell surface? Please discuss.

5) The choice of Ti3C2 delivery in the OA model was also not justified in the discussion. Why were the nanosheets injected rather than applied within a hydrogel or other formulation that kept them at the site of inflammation at the joint? What are the advantages and disadvantages to this dosing form?

6) While no cellular cytotoxicity was shown on chondrocytes due to the presence of Ti3C2 nanosheets, no discussion was given on how these agents will be excreted from the body, and whether they might cause toxicity in those excretory organs (liver, spleen, kidney). What evidence is present from literature? This must be stated as a caveat to the study that further toxicity testing is required in future studies.

Comments on the Quality of English Language

The manuscript needs to be read by a native English speaker to catch grammatical errors.

Author Response

Dear reviewer

On behalf of all the contributing authors, I would like to express our sincere appreciations of reviewers’ constructive comments concerning our manuscript. These comments are all valuable and helpful for improving our article. According to the reviewers’ comments, we have made extensive modifications to our manuscript and supplemented data to make our results convincing. In the revised version, changes to our manuscript were all highlighted within the document by using red-colored text and an underscore. Point-by-point responses to two nice reviewers are listed below this letter.

Reviewer 2

1) In the measurement of antioxidant efficiency, it would be helpful to have a positive control tested alongside the nanosheets, to give a comparator that is relatable to general literature.

Response: Thank you for the valuable comment. We add a positive control (Trolox) in the ABTS and DPPH assay to serve as a comparator. Trolox, which is a vitamin E analogue with strong antioxidant properties, is often used as positive control in ABTS scavenging assay. (Line 225-line 226)

2) The results from the osteoarthritis model require greater explanation, particularly as the negative control group (DMM) did not appear to have significant loss of cartilage structure after 8 weeks in Figure 7A. Another panel showing what undamaged, healthy cartilage looks like in the rat should be added to the figure. Also, please amend Figure 7 caption to make it clear that the two images presented for each group/staining in panel A are the same sample at different magnifications.

Response: Thank you for the valuable comment. We are sorry for lacking the healthy control. In fact, the cartilage loss in the DMM group is prominent. We replace the representative image of DMM group. The red color in the safranine O-fast green staining indicated the cartilage matrix, while the blue color in toiludine blue staining indicated the matrix of cartilage. Therefore, according to the image, the cartilage layer is much thinner than the treatment groups. In addition, the cartilage in DMM group showed a coarse surface due to heavy erosion, while the cartilage in treatment groups showed a relative smooth surface. We are sorry for failing to provide the cartilage of healthy rats now, as it takes months to decalcify the articular and obtain the staining results of the healthy control. We amend the figure 7A and caption to make it clear that the two images presented for each staining in a group are the same sample at different magnifications. (Line 317-line 318)

3) In the discussion on the mechanism of action for the nanosheets, western blotting of PI3K and AKT was mentioned, but it was not presented in the methods or data. This must be corrected, preferably with the additional data.

Response: Thank you for reminding us the incorrection, we apologize for forgetting to add the western blotting results into the supplementary material. The missing methods and data have been added. (Line 151-line 167)

4) The authors present evidence from literature for the potential benefits of nanosheets. However, there is no discussion on the reason why the authors believe Ti3C2 would be better than V2C, or a comparison to the results of the previous study using V2C. Are both of these nanosheet types believed to act via the same cellular pathway? Are they believed to be taken up by cells, or to act at the cell surface? Please discuss.

Response: Thank you for the valuable comment. We add the discussion about the therapeutic potential of Ti3C2Tx nanosheets and V2CTx nanosheets in treating OA, and discuss the cellular pathway underlying the therapeutic potential of V2CTx nanosheets. However, we are unable to conclude which one is better for OA treatment as we lack the experiments to compare their therapeutic potential in OA treatment. We believe further experiments is required to compare the therapeutic potential of various MXenes (Ti3C2Tx, V2CTx, Nb2CTx, etc.) in future studies. In addition, the engulfment of Ti3C2Tx nanosheets and V2CTx nanosheets by cell is discussed. (Line 383-line 396)

5) The choice of Ti3C2 delivery in the OA model was also not justified in the discussion. Why were the nanosheets injected rather than applied within a hydrogel or other formulation that kept them at the site of inflammation at the joint? What are the advantages and disadvantages to this dosing form?

Response: Thank you for the valuable comment. We chose to directly injected Ti3C2Tx nanosheets rather than delivered them via hydrogel as the Ti3C2Tx is unstable and could be easily oxidized, especially when they are kept in wet condition containing abundant water and oxygen. While oxidized Ti3C2Tx might be detrimental to cartilage, thus we chose to directly injected Ti3C2Tx nanosheets rather than kept them at the joint for a certain period through applying hydrogel. We extend the discussion part to discuss the dosing form. (Line 371-line 378)

6) While no cellular cytotoxicity was shown on chondrocytes due to the presence of Ti3C2 nanosheets, no discussion was given on how these agents will be excreted from the body, and whether they might cause toxicity in those excretory organs (liver, spleen, kidney). What evidence is present from literature? This must be stated as a caveat to the study that further toxicity testing is required in future studies.

Response: Thank you for the valuable comment. We add the discussion about the way by which the Ti3C2Tx nanosheets will be excreted from the body, and give the supportive literature. Also, we discuss the necessity of future studies for further biosafety assessment. (Line 397-line 407)
